# Exploring the Interplay Between Local and Regional Drivers of Distribution of a Subterranean Organism

**Stefano Mammola** [1,2,†] , **Shlomi Aharon** [3,4,†], **Merav Seifan** [3], **Yael Lubin** [3] and
**Efrat Gavish-Regev** [4,*]

1   Department of Life Sciences and Systems Biology, University of Turin, 10023 Turin, Italy
2   LIBRe – Laboratory for Integrative Biodiversity Research, Finnish Museum of Natural History,
    University of Helsinki, 00014 Helsinki, Finland
3   Mitrani Department of Desert Ecology, Swiss Institute for Dryland Environmental and Energy Research,
    Blaustein Institutes for Desert Research, Ben-Gurion University of the Negev, Sede Boqer Campus,
    Midreshet Ben-Gurion 8499000, Israel
4   The National Natural History Collections, The Hebrew University of Jerusalem, Edmond J. Safra Campus,
    Givat Ram, Jerusalem 9190401, Israel
*   Correspondence: efrat.gavish-regev@mail.huji.ac.il; Tel.: +972-54-6596250
†   Shared first authorship.

**Abstract:** Caves are excellent model systems to study the effects of abiotic factors on species distributions due to their selective conditions. Different ecological factors have been shown to affect species distribution depending on the scale of analysis, whether regional or local. The interplay between local and regional factors in explaining the spatial distribution of cave-dwelling organisms is poorly understood. Using the troglophilic subterranean spider *Artema nephilit* (Araneae: Pholcidae) as a model organism, we investigated whether similar environmental predictors drive the species distribution at these two spatial scales. At the local scale, we monitored the abundance of the spiders and measured relevant environmental features in 33 caves along the Jordan Rift Valley. We then extended the analysis to a regional scale, investigating the drivers of the distribution using species distribution models. We found that similar ecological factors determined the distribution at both local and regional scales for *A. nephilit*. At a local scale, the species was found to preferentially occupy the outermost, illuminated, and warmer sectors of caves. Similarly, mean annual temperature, annual temperature range, and solar radiation were the most important drivers of its regional distribution. By investigating these two spatial scales simultaneously, we showed that it was possible to achieve an in-depth understanding of the environmental conditions that governs subterranean species distribution.

**Keywords:** Araneae; distribution; Levant; local scale; regional scale; ecological niche modeling; subterranean troglophile

## 1. Introduction

The distribution of a species depends on ecological and evolutionary processes occurring at both local and regional scales [1,2]. Factors that may play a role at the local scale are both abiotic (e.g., climate, geology, and topography) and biotic (ecological interactions), whereas at the regional scale speciation, historical biogeography, long-distance dispersal, widespread extinction, and invasions can affect species distribution and regional species pools [3]. While regional scale processes can determine the species pool, the small-scale biotic processes and abiotic factors act as filters and determine which of these species becomes part of the community [4].

The significance of abiotic factors becomes even greater in caves and other subterranean habitats, where the selective conditions act as strong filters for the presence of species [5]. In general, the scarcity of light, space, and nutrients in caves limits the number and abundance of resident species and selects for species adapted to a narrow ecological niche [5–7]. Due to these particular conditions, spatial patterns of subterranean organisms are usually very different from those observed in surface species [8,9]. Efforts have been made to decipher the role of different cave features (for example, microclimate, nutrient, and microhabitat availability) in driving species distribution and community composition [10–13]. Some studies have focused on the broad-scale predictors of subterranean species richness and community turnover [14–21]. Yet, still little is known about global and regional macroecological patterns in subterranean biota [8,9,22].

In recent years, it has been argued that the ecological factors explaining subterranean species distribution patterns depend to a large extent on the scale of analysis considered [23–25]. In other words, certain drivers may be important in determining the microhabitat preference of a species at a local-scale, whereas the contribution of other drivers may become significant in explaining the regional species distribution [2]. Alternatively, similar factors may explain the species distribution at both scales. Only by simultaneously investigating multiple spatial scales, it is possible to fully understand the conditions governing species distributions, because each scale allows us to detect different processes. However, to the best of our knowledge, this interplay between local and regional factors in explaining the spatial patterns of subterranean organisms is poorly described (but see [23,24,26,27] for a few examples of aquatic subterranean organisms).

Spiders (Araneae) are an important group of invertebrates inhabiting the subterranean realm [28]. Across the 48 spider families having subterranean representatives [29], there are species with different levels of adaptations and affinities to the subterranean environment, from regular cave-dwellers ("troglobionts" and "troglophiles") to species occurring only sporadically in caves ("trogloxenes" and "accidentals"). Here, we focused on troglophile spiders, i.e., species forming source populations and completing their life cycle either inside or outside of caves [30].

The spider genus *Artema* Walckenaer, 1837 (Pholcidae) includes several troglophiles, and one recently discovered troglomorphic (i.e., with morphological adaptations to subterranean life) and possibly troglobiont species [31,32]. All these species have an overall Old World distribution that stretches from West and East Africa, to the Middle East and Central Asia. Little is known about the biology of these subterranean spiders. Two troglophilic species of this genus that are found in the Levant are the recently described *A. nephilit* Aharon, Huber, and Gavish-Regev, 2017 and *A. doriae* Thorell, 1881 [33,34]. The local distribution of these species contained several gaps and their habitat preferences were completely unknown. In a recent survey of caves in Israel and Palestine (West Bank), *A. nephilit* was found to be very abundant in caves distributed along the Jordan Rift Valley, while *A. doriae* was only found in a single cave during the survey [33–35]. Current knowledge of the biogeographic distribution and autecology of *A. nephilit*, as well as many other species in the genus, indicates that it is restricted to caves and crevices. The high abundance of *A. nephilit* within the study sites in the Levant and the large body size of these spiders (adults of centimetric length, easily detected in the field) opened up the possibility to use this species as a model for exploring general ecological questions. Here, we investigated which abiotic factors affect the distribution of *A. nephilit* at a local and regional scale.

We first conducted a field-study of caves potentially inhabited by this species. We then extended the analysis to a regional scale (that is, the entire known distribution of the species) to understand whether similar environmental predictors drive the species distribution at these two distinct spatial scales.

## 2. Materials and Methods

### 2.1. Study Area

The Levant is a diverse biogeographic unit composed of Northeastern African and Northwestern Arabian plates and the Eastern Mediterranean Levantine basin [36]. It is also the junction of three continents and three climate zones: Europe, Asia, and Africa, and Mediterranean, steppe, and desert climates, respectively. Major geological events resulted in these present climates and biogeographic zones of the Levant: The establishment of the Eurasian-African land bridge, drying of the Old World subtropics, the formation of the Sahara and the Syrio-Arabian Deserts, and the formation of the Afro-Syrian Rift Valley [37]. Israel and Palestine, as part of the Levant, are composed of several climatic, phytogeographical, and zoogeographical regions [38]. The phytogeographical and zoogeographical zones in Israel and Palestine are not completely congruent. While it is common to divide this region into five phytogeograpical zones, only four zoogeographical regions are commonly used for terrestrial animals. The Irano-Turanian and the Euro-Siberian phytogeograpical zones are found mainly in high mountains such as the Hermon and the Negev Mountains. Although these zones do not correspond directly to any zoogeographical region, they are both unique by their climatic conditions. The most widespread zoogeographical region in central and northern Israel, and in the Levant in general, is the Palaearctic region. The Ethiopian and Palaeoeremic regions are found in the Negev and Judean Deserts and along the Dead Sea and the Jordan Rift Valley, while Oriental elements are scattered throughout the Levant [37,38].

The Levant has a high diversity of primary caves, namely caves formed with the rocks surrounding them, and secondary caves that were formed after deposition of the rocks surrounding them. These caves differ in their rock substrate, microclimate, and age [39]. Analysis of secondary mineral deposits (speleothems) in caves in the center of Israel showed evidence of fluctuations in temperature as well as humidity regime, indicating that the climate in the Levant was either hot and wet or hot and dry [40].

### 2.2. Local Scale

#### 2.2.1. Field Survey

From 2013 to 2015, we sampled 33 caves (Table 1) over a north-south range in Israel and Palestine (West Bank) (Figure 1, enlarged map). The caves were situated in three zoogeographical regions: Ethiopian (Jordan Rift Valley and Dead Sea Valley (Israel and Palestine (West Bank))), Palaeoeremic (Negev Desert including the 'Arava Valley (Israel)) and Palearctic (central and northern Israel including the mountains of the Upper Galilee and Judea and Samaria (Israel and Palestine (West Bank))). The sampling sites covered a wide range of cave sizes (small to large caves, as defined below), entrance size, and geological substrates (limestone, basalt, clay, and conglomerate). We determined the size of each by considering the development (cave linear length) and the presence of a twilight (transition) zone and a dark zone. Large caves ($n = 12$) were caves longer than 20 m and included a twilight zone and a dark zone. Medium caves ($n = 10$) were 10–20 m with a twilight zone at the end of the cave but lacking a dark zone. Small caves ($n = 11$) had a maximum length of 10 m and lacked both a twilight zone and a dark zone.

**Table 1.** List of the 33 caves sampled in the local scale survey, arranged by size and geographic region. Letters in parentheses indicate the zoogeographical region: (P)—Palaearctic; (E)—Ethiopian; (PE)—Palaeoeremic (after [37,38]). Localities in Israel and Palestine (West Bank) and transliterated names of the localities follow the "Israel Touring Map" (1:250,000) and "List of Settlements," published by the Israel Survey, Ministry of Labor. Geographic coordinates are given in WGS84 (decimal degrees). Localities in Palestine (West Bank) are marked by asterisk (*).

| Geographic Region | Large Caves (>20 m) | Medium Caves (10–20 m) | Small Caves (<10 m) |
|---|---|---|---|
| Northern Israel | Yir'on (cave 1; 33.0679 N, 35.4665 E) (P) <br> Shetula (33.0873 N, 35.3169 E) (P) <br> Berniki (cave 1; 32.7775° N, 35.5401° E) (P) <br> Ezba' (32.7118° N, 34.9747° E) (P) | Oren (32.7144 N, 34.9749 E) (P) <br> Yonim (32.9236 N, 35.2168 E) (P) <br> Berniki (cave 2; 32.7768 N, 35.5413 E) (P) <br> Susita (32.7793 N, 35.6577 E) (P) | Yir'on (cave 2; 33.0672 N, 35.4672 E) (P) <br> Pelekh (32.9324 N, 35.238 E) (P) <br> Berniki (cave 3; 32.7775 N, 35.5401 E) (P) <br> Horvat Raqqkit (32.7128 N, 35.0123 E) (P) |
| Central Israel and Palestine (West Bank) | Haruva (31.9133 N, 34.9607 E) (P) <br> Bet 'A'rif (Shoham; 32.0026 N, 34.9642 E) (P) <br> Sali'it (32.2454 N, 35.0456 E) (P) <br> Te'omim (31.7262 N, 35.0217 E) (P) | Perat 'Inbal (cave 1)* (31.8332 N, 35.3019 E) (PE) <br> Perat Roa'im (cave 2)* (31.8325 N, 35.3130 E) (PE) | Perat (cave 3)* (31.8321 N, 35.3083 E) (PE) <br> Perat Southern Slope* (31.8334 N, 35.3054 E) (PE) <br> Oah (32.0053 N, 34.9722 E) (P) <br> Tinshemet (31.9994 N, 34.9681 E) (P) |
| Southern Israel and Palestine (West Bank) | Ashalim (30.9434 N, 34.7391 E) (PE) <br> Malcham (31.0765 N, 35.3971 E) (E) <br> 'Ammude 'Amram (cave 1; 29.6515 N, 34.9336 E) (PE) <br> Zavoa' Cave (31.2086 N, 35.2311 E) (PE) | Telalim (30.9734 N, 34.7929 E) (PE) <br> Arubotayim (31.1016 N, 35.3900 E) (E) <br> Qumeran* (31.7556 N, 35.459 E) (E) <br> Nahal Ha'Besor (30.9415 N, 34.6961 E) (PE) | 'Avedat (30.7941 N, 34.7720 E) (PE) <br> Nezirim- Ne'ot HakKikkar (30.9911 N, 35.3465 E) (E) <br> 'Ammude 'Amram (cave 2; 29.6518 N, 34.9337 E) (PE) |

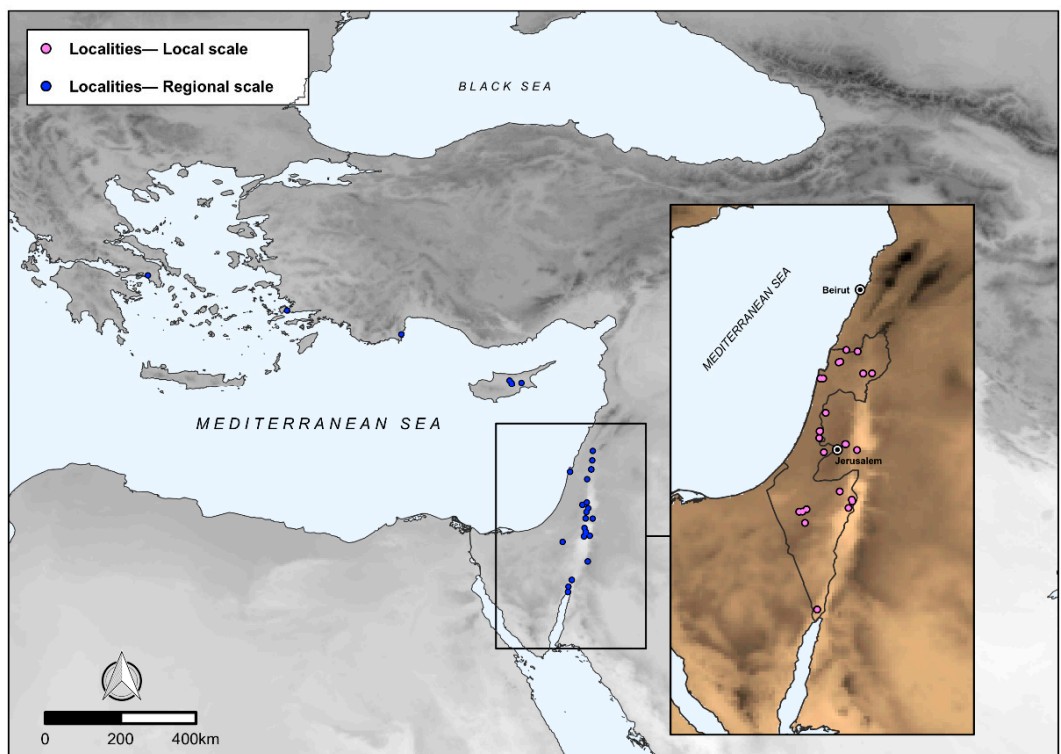

**Figure 1.** Regional distribution of *Artema nephilit* (blue dots). Local scale sites investigated in Israel and Palestine are shown in the inset map (purple dots). Background (scale of grey and brown in the main and inset map, respectively) represents elevation.

We conducted the survey in three different seasons: Late summer to autumn (August to October), spring (February to March), and late spring to early summer (April to July). With the aid of flashlights, we collected spiders by hand in four sectors: Outside the cave near the entrance, inside the cave in the vicinity of the entrance, in the intermediate part of the cave (twilight zone) and in the dark zone, when it was applicable. Each cave sector was sampled for 20 min, resulting in a minimum search time inside the cave of 20 min for small caves and a maximum search time of one hour for large caves. Adult spiders were collected and preserved in 75% ethyl alcohol. In caves where only juvenile *Artema* were found, we collected them alive and reared them in the laboratory (25 °C, 45% humidity, 12:12 L:D) until they molted to adults; then we identified them under a dissecting microscope and preserved them in 75% ethyl alcohol.

The physical and climatic attributes of each cave (length, opening size, temperature, humidity, and illuminance) were recorded. Temperature and illuminance were measured also outside the cave. Cave length was obtained from the Israel Cave Research Center, and the opening size was classified in three categories (small (<0.8 m), medium (0.8–2 m), and large (>2 m)). Temperature measurements were taken using PicoLite 16-K and a Single-trip USB Temperature Logger (FOURTEC), with measurements taken once an hour for 74–77 days. Illumination was recorded at the time of sampling using an ExTech 401025 Lux Light Meter. Elevation, precipitation, and geological data were provided by the GIS (geographic information system) center, The Hebrew University of Jerusalem. For each sampling sector, we calculated the mean temperature (T_mean), the annual temperature range (T_range) and the difference between the mean internal and mean external temperature (Delta_T). The relative humidity of each sector was expressed as a dummy variable with two levels ("dry" and "humid") depending on the presence/absence of a film of water on the substrate.

### 2.2.2. Local Scale Model

We conducted all statistical analyses in R [41]. We initially explored the dataset following the protocol for data exploration proposed by Zuur et al. [42]. We used Cleveland's dotplots to assess the presence of outliers in dependent and independent variables. We tested multicollinearity among continuous covariates via Pearson correlation tests (r), setting the threshold for collinearity at |r|> 0.7. The collinearity between continuous and categorical variables was graphically evaluated with boxplots. We excluded highly correlated predictors to avoid model overfitting.

The contributions of the measured environmental factors in explaining the abundance of *A. nephilit* spiders within all the investigated caves (*n* = 33) were analyzed at the sector level inside the caves (*n* = 69). Due to the oligotrophic conditions of the subterranean environment, the abundance of cave-dwelling spiders is often low [29]. Counts of individuals along cave transects are thus often characterized by a large proportion of zeros (that is, species absent or not detected) than expected from the pure count data [10,43,44]. In our case, more than 60% of observations were zeros and, as a consequence, Poisson generalized linear model (GLM) fitted on the data was overdispersed (overdispersion statistic = 2.94). We thus considered zero-inflated Poisson distribution (ZIP) as an alternative modeling technique rather than Poisson [45,46]. ZIPs allow modeling simultaneously the prevalence of zeros as well as the distribution of positive counts, thus resulting in two separate outputs: The first output (binomial model) explains which covariates affect the absence of individuals, while the second (count model) explains which variables drive the abundance of individuals in the occupied sectors. We used the Vuong's test [47] to compare the ZIP model versus a standard Poisson GLM, by testing the null hypothesis that both models are equally similar to the observed distribution. The test was significant (z = −1.92; *p* < 0.05), indicating that the ZIP model described the observed data more accurately than the Poisson one.

We fitted the ZIP models via the zeroinfl command in the 'pscl' R package [48,49], using a log link function for the count model and a logit link function for the binomial model. We constructed an initial model, including all covariates selected after the data exploration. We then applied a backward model selection procedure in order to simplify the initial model by removing non-significant

variables [50] according to Akaike information criterion for small sample sizes (AICc values [51,52]. This metric provides an estimation of the relative performance of regression models for a given dataset. We calculated AICc values using the 'MuMIn' R package [53]. Backward elimination process was reiterated until a minimum adequate model of fixed effects remained [50], which minimized AICc while avoiding overfitting. We tested the final model for overdispersion and validated it using residuals and fitted values [46,54].

### 2.3. Regional Scale

Once the model had been fitted using a local-scale dataset, the analysis was extended to a wider scale using species distribution models (SDMs) [55]. The aim of this analysis was to study which variables explain the current distributions of *A. nephilit* at a regional scale. The modeling procedure followed the general protocol proposed by Mammola and Leory [25] for generalizing similar spatial models to subterranean habitats. Note that we opted for different types of models to explore the spider distribution at these two scales of analyses owing to the different nature of the data we had, namely field-collected data at the cave sector level at a local scale versus geo-referenced occurrence data at the regional scale.

### 2.3.1. Occurrence Data

We assembled the regional spider distribution from the results of the field cave survey, occasional collections, historical material deposited at the National Arachnid Collection at The Hebrew University of Jerusalem, and museum and university collections around the world [33–35]. All occurrences (Figure 1) were geo-referenced (accuracy <10 m). Prior to model fitting, we designated a sampling grid at the resolution of the environmental predictors (2.5 min; see below). Rather than using raw point-locality occurrence data of *A. nephilit*, within each cell in the grid, we aggregated occurrences to avoid local overexpression of the numbers of occurrences as a result of spatial sampling heterogeneity.

### Environmental Predictors and Calibration Area

To represent the subterranean conditions across the regional species distribution, we employed both topographical and bioclimatic variables. In line with the most recent recommendations, we selected the initial set of predictors based on expert opinion [25,56,57], and only then sub-selected the final predictors using statistical inference. Prior to model fitting, we tested the multicollinearity among predictors [58], using the same procedure described for the local-scale model [54]. See Table 2 for the list of predictors used in the analysis and a discussion of their ecological relevance in representing subterranean conditions.

We calibrated and projected SDMs within the accessible area [59–61], which we defined by buffering the occurrence records by a radius of 500 km and combining all circles in a final shapefile that we used to mask the environmental predictors. We chose this relatively large accessible area to account for the fact that *Artema* spiders are widely distributed and were also collected in surface habitats [33,34,62], similar to previous SDM analyses on troglophile spiders showing high ecological plasticity and dispersal potential [63].

**Table 2.** Variables used in the species distribution models and their relevance to represent subterranean conditions. Em dash (—) indicates variables that were excluded due to collinearity.

| Variable | Source | Ecological Relevance | Permutation Importance (PI) |
|---|---|---|---|
| Solar radiation (kJ m$^{-2}$ day$^{-1}$) | [64] | Influences microclimate in superficial caves and shallow subterranean habitats [65]. Proxy for the overall habitat aridity. | 52.1 |
| Mean annual temperature (°C) | [66] | Correspond to the internal temperature of most caves [67]. | 14.8 |
| Temperature annual range (°C) | [66] | Proxies of daily and seasonal thermic excursions in the vicinity of the cave entrance [67] and in other superficial subterranean habitats [65,68]. | — |
| Daily temperature range (°C) | [66] | | 31.2 |
| Annual Precipitation (mm) | [64] | Influences general underground climatic conditions [67]. | — |
| Water vapor pressure (kPa) | [64] | Influence underground microclimatic conditions. Proxy of surface productivity [25]. | 0.3 |
| Wind speed (m s$^{-1}$) | [64] | Proxy for external energy inputs via anemochoric transportation. Air currents also influence general microclimatic conditions in certain caves [69]. | — |
| Elevation (m) | [66] | Surrogate of topographic heterogeneity and habitat availability [18,19]. It also has a general influence on climatic conditions [67]. | 0.8 |
| Carbonate substrate extent (binary raster) | World Map of Carbonate Rock Outcrops v. 3.0 | Proxy of the general availability of subterranean habitats in carbonate substrates [14,20]. | 0.7 |

### 2.3.2. Modeling Procedure

Given that reliable absence data across the whole distribution for *A. nephilit* is lacking, we fitted SDMs using a presence-only algorithm (maximum entropy model, MaxEnt; [70]). We constructed the model with the maxent function in the 'dismo' R package [71]. In consideration of the reduced sample size of our dataset (<100 occurrence), MaxEnt's feature classes and regularization multipliers were specified manually rather than using default settings [72]. We estimated the most suitable configuration of these two parameters via the ENMevaluate function in the 'ENMeval' R Package [73]. As a result of this preliminary estimation, both linear and quadratic feature classes were allowed, and a 0.5 regularization multiplier was specified.

To evaluate model performance, we ran 50 model replicates, keeping a random partition of 20% of the occurrence points for assessing model performance (evaluation dataset). We evaluated the performance on each random bootstrap partition via the Boyce index in the "ecospat" R package [74], one of the most appropriate metric with presence-only models [75]. This model validation method interprets predicted-to-expected ratio ($F_i$) by partitioning habitat suitability predictions into classes and by calculating their frequencies. A Spearman correlation coefficient was then used to estimate model fitting for $F_i$. Boyce index ranges continuously from −1 to 1; a positive value for the Spearman coefficient of $F$i denotes a model whose predictions are consistent with the distribution of presences in the evaluation dataset; values close to zero indicate that the model is not different from random, and negative values indicate counter predictions [75].

Once the models had been validated using the bootstrapping procedure, a final model was generated using the full set of occurrence points. We calculated the relative contribution of each variable to the construction of the final models via permutation importance (PI) [76], where the higher the permutation importance is, the greater the contribution of the variable.

## 3. Results

### *3.1. Local Scale*

Eighteen caves out of the 33 investigated in this study were inhabited by *A. nephilit*. The remaining caves yielded no *A. nephilit* individuals. The number of individuals of *A. nephilit* per cave ranged from 1 to 11 (mean ± s.e. = 1.30 ± 0.74). The mean number of individuals (±s.e.) per cave sector ranged from 1.61 ± 0.48 in the vicinity of the entrance to 0.36 ± 0.20 in the inner sectors.

In light of the initial data exploration, one outlier illuminance value was removed from the dataset and a log-transformation was applied to illuminance to homogenize its distribution. The exploratory analysis, whose goal was to select potential abiotic variables for the model, revealed that internal temperature range was collinear with mean internal temperature and log-transformed illuminance (both r > 0.7). Therefore, this variable was excluded from the analysis. The categorical variables referring to the cave size and opening size were also excluded from the model, being collinear with mean internal temperature and illuminance, respectively. As a result, the initial model included the variables mean and delta temperature, relative humidity and log-transformed illuminance.

According to the model selection (Table 3), the most parsimonious ZIP model explaining the abundance of *A. nephilit* (count model) and its absence (binomial model) in the investigated caves included log-transformed illuminance and mean internal temperature. Overdispersion in the final model was minimal (dispersion statistic = 1.34). The abundance of the species was found to increase at increasing mean internal temperature values (count model, estimated β ± s.e. = 0.11 ± 0.05, *p* < 0.05) and increasing illuminance (count model, estimated β ± s.e.= 0.15 ± 0.08, *p* < 0.05). Mean low internal temperature also significantly decreased the probability of finding *A. nephilit* in the cave sector (binomial model, estimated β ± s.e. = –0.76 ± 0.23, *p* < 0.001), whereas no significant effect was detected with respect to log-transformed illuminance (binomial model, estimated β ± s.e. = –0.25 ± 0.28, *p* = 0.37). The predicted conjunct effect of these two continuous variables on the abundance of *A. nephilit* in the cave sectors is illustrated in Figure 2.

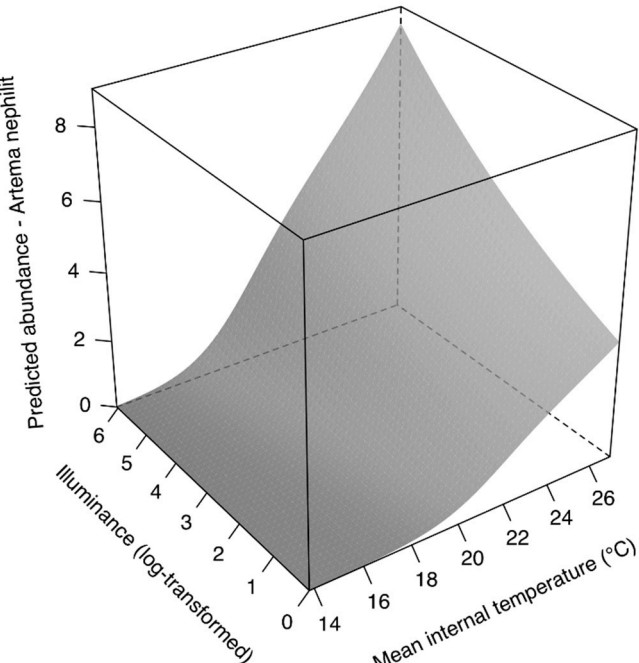

**Figure 2.** Predicted variation in abundance of *Artema nephilit* along two environmental gradients within the cave (mean temperature and illuminance) according to ZIP analysis. The variable illuminance was log-transformed to homogenize its distribution.

**Table 3.** Model selection according to Akaike information criterion corrected for small sample size (AICc; [51,52]). Models are ordered from the most to the less appropriate. Df = degrees of freedom; AICc = corrected Akaike information criterion for finite sample size [53]; ΔAICc = (AICc of the model) − (AICc of the best model); wi = Akaike weight [52]. Abbreviations of the explanatory variables are in the main text.

| Model Structure | Df | AICc | ΔAICc | wi |
|---|---|---|---|---|
| y~ T_mean + Illuminance | 6 | 172.97 | 0.00 | 0.73 |
| y~ T_mean + Delta_T + Illuminance | 8 | 175.31 | 2.33 | 0.22 |
| y~ T_mean + Delta_T + Humidity + Illuminance | 10 | 178.65 | 5.68 | 0.04 |

*3.2. Regional Scale*

We included in the SDM analysis occurrence records from the entire known range of distribution of the species. Overall, we assembled 64 unique geo-referenced records for *A. nephilit*. As the result of the sampling grid correction, occurrences were filtered down to 27 occurrence records.

According to the standard validation metrics, the SDM predictive performance for *A. nephilit* was adequate (Boyce index; mean of 50 bootstrap replicates ± s.e.: 0.64 ± 0.07). The contribution of the environmental predictors in determining the distribution is presented in Table 2. Over 80% of the modeled distribution for *A. nephilit* was explained by solar radiation (PI = 52.1), daily temperature range (PI = 31.2), and mean annual temperature (PI = 14.8). The contribution of the remaining predictors was limited (all PI < 1).

Suitable areas predicted by the model mostly overlaid the known distribution of *A. nephilit* (Figure 3). However, the model largely failed to predict the presence of the species in Greece. The model also predicted suitable areas for the species in a few areas in which the species have never been previously recorded. Specifically, the model predicted occurrence in Egypt and Saudi Arabia, and in Turkey all along coastal areas facing the Mediterranean Sea.

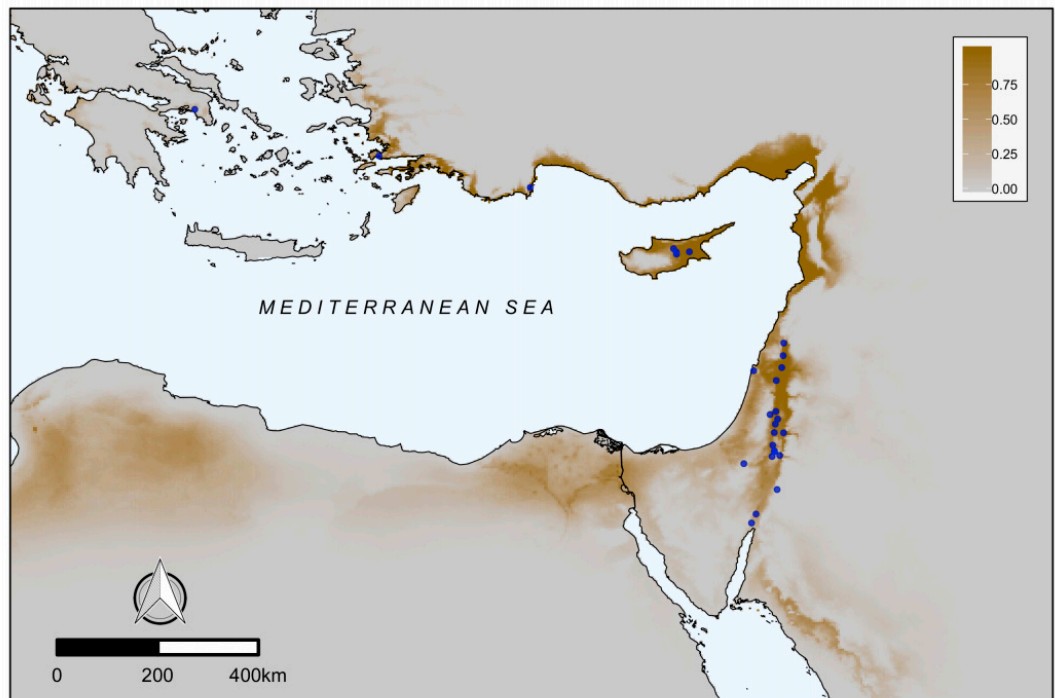

**Figure 3.** Current distribution map for *Artema nephilit*. Colored scale gradient represents the probability of presence of the species based on a species distribution model. Blue dots are the presence records of the species used in the model.

## 4. Discussion

### 4.1. Models Interpretation

Prior to this study no information was available about the ecological requirements of *Artema* spiders, aside from anecdotal records of habitat preference found in the taxonomic literature [33,34,62]. The goal of this study was thus to investigate the ecological factors determining the distribution of the large troglophile spider *A. nephilit* at local (microhabitats in Israel) and regional (whole distribution) spatial scales. The two strongest and significant predictors of the abundance of *A. nephilit* at the local scale were illuminance and mean annual temperature inside the cave (Table 3). Specifically, the predicted abundance of individuals was higher in the most illuminated sectors of the caves (Figure 2), which are typically found in the vicinity of the entrance. The selection of the most external areas of the cave is commonly observed in non-troglobiont species (e.g., [11,77–80]) and is consistent with the relatively weak degree of subterranean adaptation of this species. Transitional habitats at the surface/subterranean interface are known to be the richest in term of abundance and diversity of species [81,82], which indicates that the entrance area is potentially the richest cave sector in term of available prey for *Artema*. In this respect, it is worth noting that the variable illuminance was also positively correlated with the opening size, i.e., the larger the opening, the more light reaches the cave. In turn, a larger opening may facilitate the occasional colonization of the cave by external organisms [83,84], which may serve as prey. *Artema nephilit* was also more abundant in warmer sectors of the cave (Figure 2), which is consistent with its apparent thermophily. Indeed, observations that were made during the cave survey, as well as the result of the SDM analysis (see below), indicates that *A. nephilit* prefers hot and relatively dry caves rather than cool and humid caves that are the preferred habitat of other cave-dwelling pholcid spiders found in the surveyed caves [31,33]. In areas where both hot and dry caves and cool and humid caves were present, *A. nephilit* inhabited the hot dry caves exclusively, while other pholcid spiders inhabited the cool humid caves [33].

At the regional scale, the ecological predictors driving the distribution of *A. nephilit* were very similar to the factors recovered in the local scale (Table 3; Figure 2). The congruence between relevant abiotic factors at the two scales suggests that *A. nephilit* is conservative in its ecological niche across the distribution range. It is worth noting that our SDM analysis was based on a relatively low number of occurrences, which may explain why the model failed to predict occupied areas at the margins of the distribution in Greece. Surprisingly, this geographically widely distributed species is documented from relatively few localities. As noted by Aharon et al. [34], most diversity in the genus *Artema* is concentrated in a region of the world that has seen much political instability recently. Given this perspective, the distribution map provided here (Figure 3) might be useful both for filling knowledge gaps and for identifying suitable sites for further sampling.

### 4.2. Biogeographic Considerations

The current distribution of a species can be explained by two contrasting hypotheses using historical-biogeography: Vicariance and dispersal. Vicariance is the fragmentation of widespread ancestors by isolating events, creating disjunct distributions, relicts or even allopatric speciation [85]. By contrast, long-distance dispersal among regions is a process that enables individuals to colonize new sites, thereby reducing fragmentation and increasing the distribution range [86–88]. Both fragmentation and dispersal processes are also likely to be influenced by changing climates, which may lead to extreme changes in distribution due to extinctions and novel invasions [89,90].

Initially we assumed the distribution of *A. nephilit* to be a classic Ethiopian or Palaeoeremic. However, the field cave survey, occasional collections, material deposited in collections [33,34], and the results of this study, indicate that *A. nephilit* distribution in Israel is Eastern-Mediterranean, as reflected in its presence in Greece, Cyprus, and Turkey and its absence from African regions. While most of the sites in which *A. nephilit* was found were associated with the Rift Valley, two sites (Avedat and Oren) were outside the Rift Valley. These two sites in fact are connected by valleys that could

provide passageways from the Rift Valley (Avedat by Zin Wadi and Oren via the Yizre'el Valley). The preference of *A. nephilit* for dry and eastern sites in the local-scale cave survey and the regional-scale distribution noted above lead us to raise two different hypotheses about the species' distribution in the Mediterranean basin.

The long-distance dispersal hypothesis suggests that *A. nephilit* originated in the Jordan Rift Valley and expanded its distribution to suitable habitats and other geographical regions using ecological corridors. For example, the Yizre'el Valley may have been a suitable corridor expanding the distribution to Turkey, Cyprus, and Greece to the northwest, and the Moab and the Edom canyons could explain the current distribution in Edom Mountains in Jordan. Specific surveys of caves along these proposed corridors and a molecular analysis of individuals from different regions might help to test this hypothesis.

The vicariance hypothesis suggests that *A. nephilit* had a wider Mediterranean distribution during a warmer climatic period; following historical climate changes, the species distribution narrowed to suitable habitats within its preferred hot and dry climate. This resulted in isolated populations within its original distribution area and relicts in suitable habitats. This hypothesis of a climatic-relict distribution is often adopted to explain the colonization of the subterranean environment [91]. However, most examples are of relict distributions in caves located in cool, humid habitats following the retreat of glaciers ([92] but see [93]). As for spiders, it is suggested that glaciation cycles played a major role in determining the pattern of subterranean spider biodiversity in Europe (reviewed in [94]). It is less clear what climatic factors would result in a relict distribution in dry, warm caves (although aridification has been shown to be a potential driver of subterranean diversification [95,96]), especially when there is no evidence of an afrotropical origin. Yet, recently it was suggested that a troglobitic harvestmen (Opiliones: Laniatores) from a humid and warm cave in Israel is an afrotropical relict, although it belongs to a genus (*Haasus*, Roewer, 1949) that was previously suggested to be a palearctic relict [97].

## 5. Conclusions and Perspectives

The combination of a thorough field survey and distribution analysis at two scales—local and regional—enabled us to reveal the main factors driving the local distribution of *A. nephilit*, and to shed light on its biogeography and ecological requirements across its regional distribution. Interestingly, the analyses recovered similar ecological factors determining the distribution at both the local and regional levels for *A. nephilit*. Overall, these results represent the first quantitative description of the autoecology and biogeography of this spider species.

This study also provides baseline information for future analysis of the distribution of the other species of *Artema*. Given the availability of an extensive spatial dataset for the genus *Artema* after the recent revision by Aharon et al. [34], there is a strong potential for extending the biogeographic analysis to the eight nominal species of the genus. In this respect, combining SDM projections with genetic data at the population/species interface would allow an in-depth study of the past distribution of these spiders, their evolutionary history in surface and subterranean habitats and their potential future response to environmental alteration due to climate change.

**Author Contributions:** Conceptualization: S.A., Y.L. and E.G.R.; Methodology: S.A, S.M., M.S., Y.L. and E.G.-R.; Analysis: S.M., S.A., M.S., Y.L., E.G.-R.; Investigation: S.A. and E.G.-R.; Resources: E.G.-R., Y.L. and S.M.; Data Curation: S.A. and E.G.-R.; Writing—original draft preparation: S.A., S.M. and E.G.-R.; Writing—review and editing: S.A., S.M., M.S., Y.L. and E.G.-R.; Project administration: E.G.-R.; Visualization, S.M. and E.G.-R.; Supervision, E.G.-R., M.S., and Y.L.; Funding acquisition: E.G.-R. and Y.L.

**Funding:** This research was funded by the Israel Taxonomy Initiative (ITI) biodiversity survey grant, and ITI and Ben-Gurion University fellowships to S.A.

**Acknowledgments:** The research was conducted with collection permits 2013/40027, 2013/40085 and 2014/40313 from the Israel National Parks Authority. We thank M. Isaia and R. Kadmon for discussions on the arachnid cave survey and this manuscript; B.A. Huber for taxonomical collaboration; B. Langford and E. Cohen (Israel Cave Research Center); and I. Armiach-Steinpress (Israel National Arachnid Collection) for assistance in the field and for illustration. This is publication number 1034 of the Mitrani Department of Desert Ecology.

**Conflicts of Interest:** The authors declare no conflict of interest.

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
