# Peer review of "Exploring the Interplay Between Local and Regional Drivers of Distribution of a Subterranean Organism"

_diversity, doi:10.3390/d11080119_

Reviewer 1 Report

I enjoyed reading the mauscript. I congratulate the authors for their well done statistical analysis. Please consider my few comments on the text and style.

Abstract

L20: substitute „we asked“ with „we investigated“

L25: A. nephilit cursive

Introduction

L35: occurring at both ….

L41: I have problems with the word “harsh” because species are adapted to the conditions and cannot drive in other environments; of course, no light is harsh but when you are used to it then you cannot live somewhere else. Thus, I suggest deleting “harsh”

L54: other drivers

L74: caves in Israel,

L77: please remind, results indicate and humans suggest; thus, please write: “… in the genus, indicates that it is …..”

L77-80: no comma needed in the whole sentence

L80: please write: We investigated which abiotic factors affect the distribution of A. nephilit at a local and global scale.

L82: “in order to” should be shortened to “to”

L85: … we investigated whether …

Material and Methods

L106: high diversity

L110: indicating

L114: From 2013 to 2015, …

L118: please write: “The sampling sites covered a wide range of cave sizes …..”

L149: … from the Israel Cave Research Center (ICRC), …..

L155: For each sampling sector, we ….

L188: please add more information regarding model reduction; I guess that what you did is model averaging.

L202: please write: “All occurrences (Figure 1) were geo-referenced (accuracy<10 m).”

L203: records from …

L223: indicates

Results

L249-255: while I find it positive that you explain why you excluded variables from the model, now I do not know which variables entered the model; please, add just one sentence that enumerates variables used.

L258: delete “as independent variables” because redundant considering the method section and the sentences before

L281: A. nephilit

Discussion

L304: delete “of analysis”

L311/326: indicates

L355: .. to the north-west, ….

L357: no comma after “corridors”; furthermore, please write “proposed” instead of “suggested”

L360: split the sentence in two; I suggest putting a semicolon after “warmer climatic period”

Author Response

We corrected all suggested typos in the text and the reference list, Please see the attachment.

Reviewer 2 Report

This manuscript focuses on identifying the environmental factors that determine the distribution of the pholcid spider Artema nephilit linking measured microhabitat conditions at a local scale with region-wide environmental conditions. The authors use a combination of modeling approaches to show that similar ecological factors help define the distribution of the spider at local and regional scales. 

The authors did an excellent job in submitting a manuscript that is very well written and nearly void of grammatical mistakes. They presented their findings in a clear and descriptive way that does not overstate or extend beyond what their data reveals. The results seem solid based on the analyses they made. Overall, I think this is a really neat paper! I only have two minor concerns regarding some statements in their interpretation of the models, and a few suggestion points that the authors should consider to improve the manuscript.

Concern 1:
The title of this study boasts about the “interplay between local and global drivers of distribution” but the study organism is fairly narrowly distributed. Please consider the use of “regional” instead of “global”. In turn, please use “regional” throughout the text in places where a “global scale” is referred.

Concern 2: 

In line 323, the authors claim “… we produced an SDM projection that convincingly represent[s] the species range.” The use of “convincingly represent[s]” has two problems in this context: 1. Please remember that SDMs only help us identify the suitable environmental areas where species can have viable populations, and in this case, this is done using a correlative modeling approach of abiotic conditions. Thus, the projected distribution can only approximate the species range because other factors are not accounted for. 2. “convincingly” is a qualitative adverb that has no real statistical meaning. Adding a quantitative measure, even something as simple as true positive fraction, would give you a quantitative edge that measures how good was the fit of the model to the known present distribution (i.e. localities) where the species occurs. For example, if your model correctly predicts 95% of your species localities, then you can argue that this is a reasonable estimate of where suitable environmental conditions for this species occur (as opposed to a model that may only correctly predict, say, 50% of the localities).

Minor points:

Introduction

L65: why is “Accidentals” capitalized?

L65: “… we focused on troglophile spiders …”: so, would these be “trogloxenes”? Perhaps define in which category in which “troglophiles” are classified.

L71: While the Levant area is described in appropriate detail later, here is seems to come out of the blue with no description. Perhaps add a brief reference to what Levant is for those who are not familiar with it.

L72: “… survey of caves in Israel A. nephilit was …”: Please add a comma after Israel

Materials and Methods

L137: Please correct the “degree” symbol

L141: Change “indicates” with “indicate”

L150: Does it make sense to have an entrance size of zero in the “small” category?

L211: “… initial set of predictors on an expert-based basis …”: Perhaps change to “… initial set of predictors based on expert opinion …”

Results

L247 and other instances throughout: “… (mean±s.e.=1.30±0.74) …”: Please use spaces to make these readable.

L289: As suggested above, ideally, you could estimate a True Positive Fraction to support this statement.

Figures

Figure 2: Please change the axis label of the figure “Light intensity” to “Illuminance” to improve clarity and consistency

Figure 3: In general, the geographic extent in Figure 3 seems a bit broad. Perhaps a zoomed in version covering a smaller extent and trimming part of the surrounding non-suitable areas would be easier to read. Also, along the lines with the suggestion for L289, Figure 3 could be improved by also adding the locality points where the species is known to occur. This will provide a clear visualization of how well the predicted areas fit the locality data used for modeling. Finally, in Figure 3 caption: Does this truly represent the “probability of presence of the organism”? Or does it represent the areas of suitable conditions where the species can sustain viable populations? Carefully consider revising this.

Discussion

L323: See concern 2 listed above and carefully consider revising this statement.

L327” See notes on Figure 3 to help improve the visualization and clarification of this statement.

Conclusions and future perspectives

Consider adding a broader perspective reflecting the importance of this spider group to highlight the necessity to understand their biology. 

References:
Please carefully revise all references for proper formatting.

Author Response

Please see the attachment including most of the minor corrections suggested by the reviewer.

Reviewer 3 Report

Only minor suggestions. Great job, 

Author Response

Reviewer 3 asked regarding the Title
"Should it be among instead of between? Between is only for comparing two things, among for multiple"

It is “between” because it is between local and global drivers – 2 items (even though there might be several of each).

We Corrected all suggested typos in the text and the reference list; We included toponyms in figure 1 and 3 to allow a reader to orientate more easily.

Note that we cannot provide source to the map in the caption of figure 1, as this is an original figure prepared by the authors using a standard elevation layers and the spider’ distribution points;

We changed colour of figure 3 to avoid losing information in case of B/W conversion;
